# Rheological Properties of Composite Inorganic Micropowder Asphalt Mastic

Tengteng Guo [1], Haijun Chen [1], Deqing Tang [1], Shengquan Ding [1], Chaohui Wang [2], Decai Wang [1], Yuanzhao Chen [1],* and Zhenxia Li [1]

[1] School of Civil Engineering and Communication, North China University of Water Resources and Electric Power, Zhengzhou 450045, China; guotth@ncwu.edu.cn (T.G.); chenhaijun@ncwu.edu.cn (H.C.); tdq18738952643@163.com (D.T.); dsqcad@163.com (S.D.); wangdecai@ncwuedu.cn (D.W.); zhenxiali2009@ncwu.edu.cn (Z.L.)

[2] School of Highway, Chang'an University, Xi'an 710064, China; wchh0205@chd.edu.cn

*   Correspondence: cyz740513@ncwu.edu.cn

**Abstract:** Graphene Tourmaline Composite Micropowder (hereinafter referred to as GTCM) modified asphalt was prepared by the ball milling method. The effects of different temperatures and different frequencies on the high-temperature performance of composite-modified asphalt were evaluated by dynamic shear rheological test, and the viscoelastic properties of composite-modified asphalt under different stresses and different temperatures were analyzed. The low-temperature rheological properties of GTCM-modified asphalt were analyzed by bending beam rheological test, and its mechanism was analyzed by Fourier transform infrared spectroscopy (FTIR) test. The results show that the temperature sensitivity and anti-aging resistance of GTCM-modified asphalt are significantly higher than that of tourmaline-modified asphalt. The improvement effect gradually increases with the increase in graphene powder content, and its addition does not change the viscoelastic properties of asphalt. The complex shear modulus and phase angle of GTCM-modified asphalt at appropriate temperatures are more conducive to tourmaline-modified asphalt and matrix asphalt, which can improve the rutting resistance of asphalt. In the same type, with the increase in composite modified micropowder content, the rutting resistance of modified asphalt is better. The improvement of rutting resistance of GTCM-0.5, GTCM-1.0 and GTCM-1.5-modified asphalt can reach 12.95%, 10.12% and 24.25%, respectively; the improvement range is more complicated due to temperature and frequency changes. The GTCM-modified asphalt has good low-temperature crack resistance. The creep stiffness modulus of GTCM-modified asphalt decreases with the increase in load time under different types and dosages, and its stiffness modulus is smaller than that of tourmaline-modified asphalt and mineral powder asphalt mastic. The creep rate increases with the extension of load time, which is greater than that of tourmaline-modified asphalt and mineral powder asphalt mastic. When the load was 60 s, the creep stiffness modulus of GTCM-0.5, GTCM-1.0 and GTCM-1.5-modified asphalt decreased by 5.75%, 6.97% and 13.73%, respectively, and the creep rate increased by 1.37%, 2.52% and 4.35%, respectively. After adding GTCM or tourmaline to the matrix asphalt, no new functional groups were produced due to the chemical reaction with the asphalt.

**Keywords:** pavement material; tourmaline; graphene; asphalt mastic; rheological properties





## 1. Introduction

The overload problem of asphalt pavement is escalating in severity due to the nation's fast growth in car ownership. It is difficult for matrix asphalt to solve the problem of the long fatigue life of the pavement. However, the performance of traditional modified asphalt has gradually failed to meet the current requirements of asphalt pavement, resulting in frequent diseases of asphalt pavement and the actual service life cannot reach the design life, resulting in a serious waste of resources and economic losses [1–3]. At present, SBS

modified asphalt and crumb rubber modified asphalt are the most widely used types of modified asphalt. Due to their inherent component incompatibility, easy oxidation degradation and density difference, there are still some inevitable problems such as low road performance, poor durability, and insufficient stability [4,5], which seriously restrict the service life of pavement laid with such materials. Therefore, it is urgent to develop high-performance and long-life asphalt pavement materials.

Due to its unique material properties, the modification of asphalt by inorganic compounds has emerged as a research hotspot in the field of asphalt modification in light of the issues with conventionally treated asphalt. wide sources, and simple production equipment. To a certain extent, inorganic ingredients can enhance the overall performance of asphalt mastic and asphalt mixture, hence enhancing the quality of asphalt pavement [6]. Compared with organic-modified asphalt, inorganic powder-modified asphalt has the characteristics of a simple production process, low price, and excellent performance. Therefore, many scholars at home and abroad have carried out a series of studies on inorganic powder-modified asphalt. The commonly used inorganic micropowder materials are carbon black, fiber, diatomite, cement, hydrated lime, silica fume, layered silicate, nano-calcium carbonate, etc. Inorganic micropowder-modified asphalt is by adding inorganic micropowder materials into asphalt, through physical adsorption and chemical reaction, etc., to improve the road performance of asphalt and asphalt mixture, improve the performance of asphalt pavement, and improve the performance of asphalt mastic to a certain extent.

Graphene has been widely used in composite materials, nano-electronic devices, catalyst carriers, sensors, and energy storage due to its excellent thermal conductivity, high strength and large specific surface area [7–10]. The existing results show that the high-temperature rutting resistance and high-temperature stability of asphalt can be improved by means of the high thermal conductivity, high specific surface area and excellent mechanical properties of graphene [11–15]. Amirkhanian S et al. [16,17] studied the interaction between graphene oxide and asphalt and its road performance. The results show that graphene oxide does not chemically react with asphalt, but only a simple physical blending, but the adhesion between them is good. Graphene oxide-modified asphalt will reduce the ductility of asphalt and increase the viscosity. It significantly enhances the rutting resistance of modified asphalt within the range of 30 to 80 °C. M. Marasteanu et al. [18,19] studied the performance changes of graphene-modified asphalt before and after aging. Graphene modifiers can lessen asphalt's aging rate, postpone the production of carbonyl and sulfoxide groups during aging, and increase the resilience of asphalt to fatigue cracking. Christiansen S. [20] proposed a method for graphene to enhance the strength of foamed asphalt, indicating that graphene oxide can improve the mechanical properties of foamed asphalt. Wang Chaohui et al. [21–24] systematically studied the effects of tourmaline on the high-temperature performance, low-temperature crack resistance, water temperature qualitative and fatigue resistance of asphalt and its mixture, as well as the functions of thermal companion emission reduction, flame retardant smoke suppression and air purification. The results show that tourmaline has a good improvement effect on asphalt pavement performance and environmental efficacy. Haibo Ding et al. [25] prepared polymer-modified asphalt containing different proportions of tourmaline modifier and found that a small amount of modifier could improve the mechanical properties of asphalt binder, and the modification effect of tourmaline ion powder was the best. Kim S. J. et al. [26] prepared a $TiO_2$ photoanode with different content of tourmaline by using the spontaneous polarization characteristics of tourmaline and compared it with the original $TiO_2$ photoanode by electrochemical impedance spectroscopy, current density, and voltage. Compared with the original $TiO_2$ photoanode, the electron lifetime and power conversion efficiency of the $TiO_2$ photoanode with 3% tourmaline were increased by about 42% and 20%, respectively. Kang S. J. et al. [27] added tourmaline particles into nylon fibers doped with $TiO_2$ particles and found that tourmaline can play a synergistic role with $TiO_2$ to improve the ability of photocatalytic degradation of organic pollutants. Moreno-Navarro et al. [28] tested the rheological and thermal properties of binders prepared with different amounts of graphene

sheets. It was found that the addition of graphene to asphalt can produce a greater elastic response. The impact of tourmaline content on the photocatalytic activity of TiO2-graphene was investigated by Baeissa et al. [29]. Tourmaline's spontaneous polarization effect can enhance photocatalytic activity, and the composite material with the highest performance has a G-to-T ratio of 1% to 2.5%.

In summary, although graphene and tourmaline are two environmentally friendly materials that belong to different substances, they can exert excellent adsorption capacity and play a major role in environmental protection. When used alone as modifiers, they can greatly enhance the asphalt's rheological characteristics and fatigue resistance. However, there are few studies on the combination of the two in the road field. Therefore, this paper prepared GTCM asphalt mastic by graphene/tourmaline, and systematically studied the rheological properties of inorganic micropowder asphalt mastic, to produce composite asphalt-modified materials with cheap cost, environmental protection, excellent quality, and long life.

## 2. Materials and Tests

### 2.1. Materials

#### 2.1.1. GTCM

The composite inorganic powder refers to a powder mixture made by ball milling of graphene and tourmaline. Among them, graphene (Figure 1) is provided by Suzhou Carbonfeng Graphene Technology Co., Ltd. (Suzhou, China). The main physical properties are shown in Table 1.

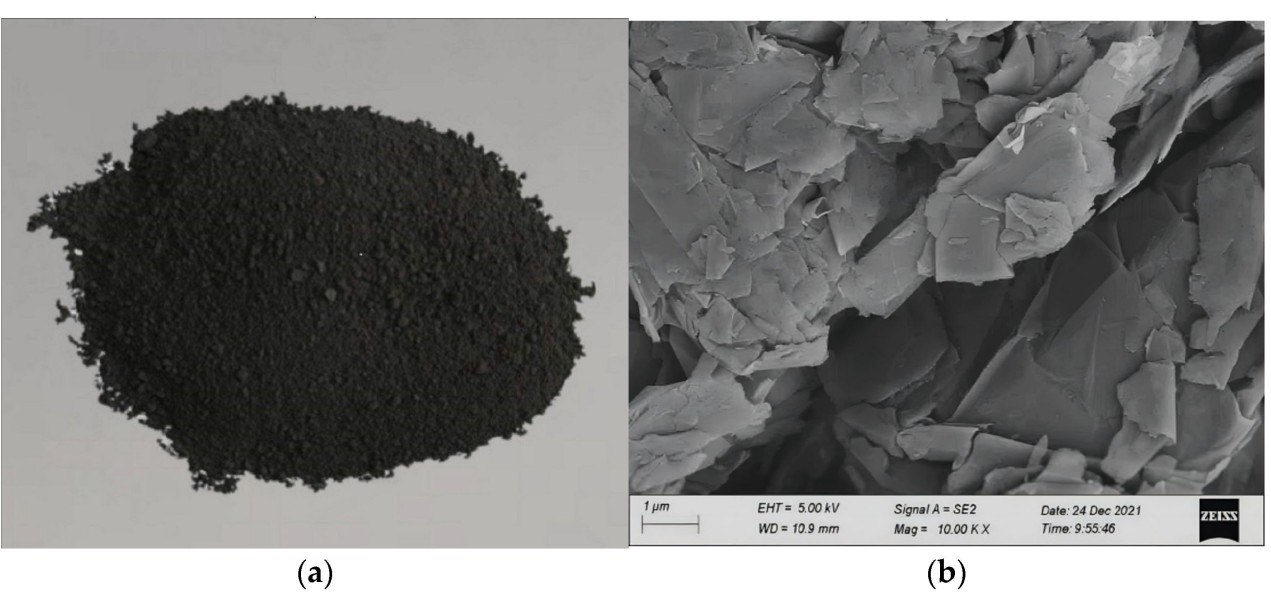

(**a**)        (**b**)

**Figure 1.** Graphene. (**a**) photo. (**b**) SEM picture.

**Table 1.** Technical specifications of graphene.

| Performance Parameter | Appearance | Purity (wt%) | Thickness (nm) | Specific Surface Area (m$^2$/g) | Layer Diameter (μm) |
|---|---|---|---|---|---|
| measured value | black powder | 95 | 1.35 | 89 | 47 |

The appearance of the tourmaline used is a black powder (Figure 2), 2000 mesh, and the main components are shown in Table 2.

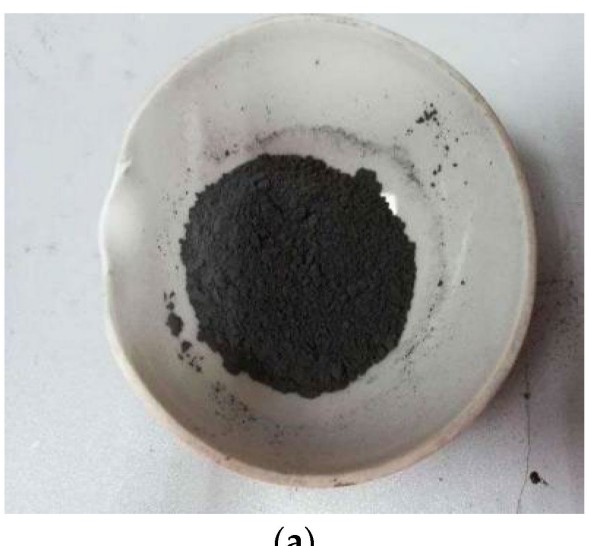
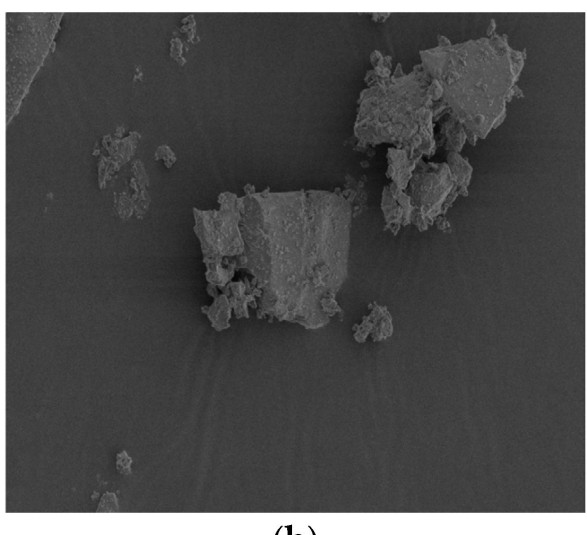

**(a)**                                                      **(b)**

**Figure 2.** Tourmaline. (**a**) photo. (**b**) SEM picture.

**Table 2.** The main components and content of tourmaline.

| Essential Component | SiO$_2$ | Al$_2$O$_3$ | B$_2$O$_3$ | MgO | FeO | Fe$_2$O$_3$ | CaO | TiO$_2$ | Na$_2$O | H$_2$O |
|---|---|---|---|---|---|---|---|---|---|---|
| content (%) | 33.54 | 31.98 | 11.21 | 0.49 | 2.88 | 16.32 | 0.01 | 0.28 | 0.72 | 2.51 |

GTCM (Figure 3) is based on tourmaline as the main material. By adding a certain mass fraction of graphene to the tourmaline powder material, the two are combined with a certain process to form a composite material with a certain process. It is expected that the two will play a synergistic role and play a synergistic role in enhancing the performance of asphalt and environmental efficacy. In view of the high price of graphene with superior quality sold in the market, the amount of graphene should be appropriately controlled when preparing the composite material. The ball milling method is used to obtain [30], in which the mass ratio of graphene to tourmaline is 0.5%, 1.0% and 1.5%, respectively. They are denoted as GTCM-0.5, GTCM-1.0 and GTCM-1.5, respectively.

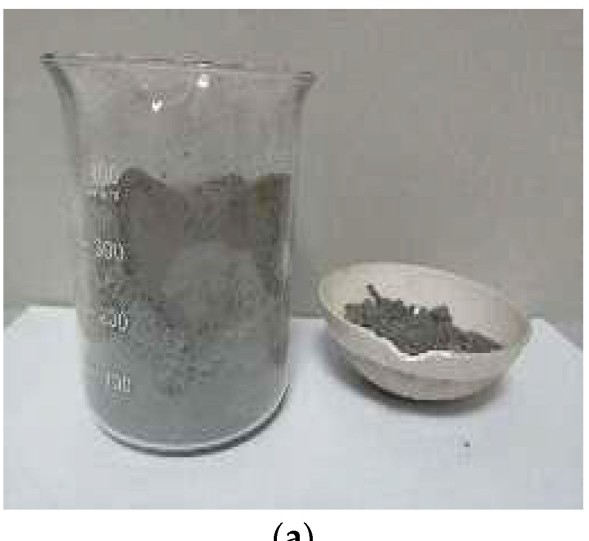
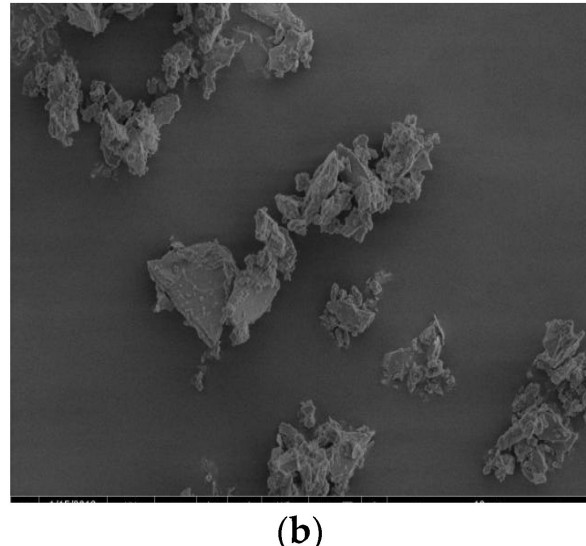

**(a)**                                                      **(b)**

**Figure 3.** GTCM. (**a**) photo. (**b**) SEM picture.

2.1.2. Asphalt

AH-70 # asphalt is used as asphalt, and Table 1 displays its technical indications.

### 2.2. Preparation of GTCM Asphalt Mastic

The performance of modified asphalt is directly impacted by the preparation procedure used to create the inorganic micropowder modified asphalt mastic. To ensure the dispersion of inorganic powder, modified asphalt was prepared using the following method:

(1) Weigh the dried asphalt after quantitative dehydration. Based on the quality of asphalt and the content of GTCM, titanate coupling agent TC-131 and quantitative dry GTCM were weighed and well combined. When the asphalt is heated to good fluidity, the uniformly mixed powder material is poured into it, and one side is added for manual stirring for 5 min.

(2) The high-speed shearing instrument was started. First, after shearing and dispersing at 1000 rpm for ten minutes, the speed was increased to 3000 rpm for thirty minutes. The rotor was taken out of the mixing barrel by adjusting the speed, and the temperature of asphalt in the modification process was controlled at 150 °C.

(3) To get rid of the air bubbles in the GTCM-modified asphalt, it was manually stirred with a stirring rod for 10 min before being placed in storage.

### 2.3. Basic Performance Test Method

According to the relevant provisions of "Test Specifications for Asphalt and Asphalt Mixtures for Road Construction" (JTG E20-2011), the penetration index (PI) and equivalent softening point ($T_{800}$) of asphalt were also evaluated. The penetration, softening point, and ductility of asphalt were examined. Asphalt was aged using the Rotating Film Oven Heating Test (RTFOT) (Ai Yao Scientific Instrument Co., Ltd., Shanghai, China) according to ASTM D2872. Aged asphalt was tested for penetration and softening point, and the residual penetration ratio and softness point increase were computed.

### 2.4. Test Method for Rheological Properties

2.4.1. Test Method for Dynamic Shear Rheological Properties

The H-PTD200 dynamic shear rheometer was produced by Anton Paar (Graz, Austria), as shown in Figure 4. The temperature control range of the instrument is −40~+200 °C, supporting the speed control range of $10^{-6}$ μrad/s~314 rad/s, equipped with a variety of parallel plates and fixtures, and supporting the test of asphalt samples under various modes such as temperature scanning and frequency scanning.

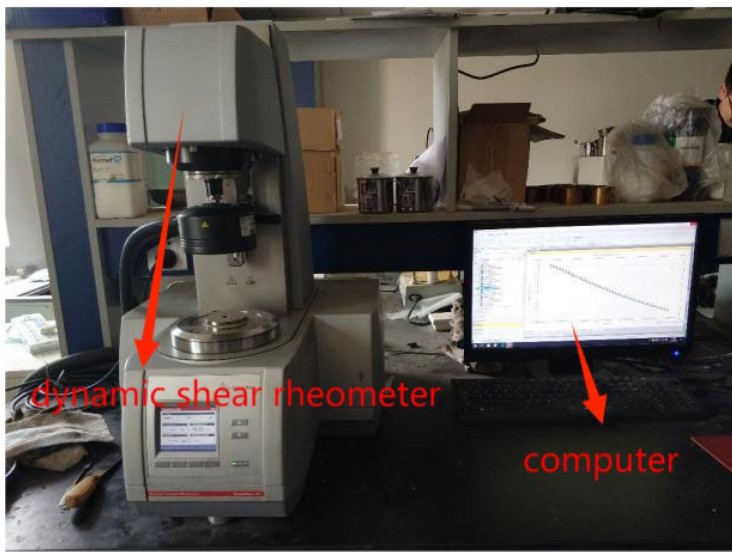

**Figure 4.** Dynamic shear rheometer.

In the experiment, the specimen was removed from the mold and placed between two parallel plates with either an 8 or 25 mm diameter. The distance between the plates was adjusted to meet the experimental requirements. A scraper was used to remove extra asphalt outside the plates. The upper plate spun around the central axis at a predetermined angular rate while the lower square circular plate remained fixed during the test. A sensor recorded the test results.

Dynamic shear rheological tests were conducted on GTCM-modified asphalt at varying temperatures and frequencies, with different types and dosages. The resulting data allowed for a comprehensive and objective evaluation of the dynamic shear rheological properties of the modified asphalt. Table 3 shows the specific test plan which includes a control strain of 1% and a frequency of 10 rad/s, with a temperature range of 30~80 °C. Additionally, a frequency scanning control strain of 10% will be used with a temperature of 60 °C and a frequency range of 0.1~100 rad/s.

**Table 3.** DSR test scheme of GTCM asphalt mastic.

| Order Number | Types | Dosage (%) |
|:---:|:---:|:---:|
| 1 | asphalt | - |
| 2 | T asphalt mastic | 20 |
| 3 | GTCM-0.5 asphalt mastic | 20 |
| 4 | GTCM-1.0 asphalt mastic | 10; 20; 30 |
| 5 | GTCM-1.5 asphalt mastic | 20 |

2.4.2. Low-Temperature Rheological Property Test Method

The TE-BBR-F type low temperature bending beam rheometer produced by CANNON company (Peschiera Borromeo (MI), Italy) was used. As shown in Figure 5, according to the ASTM D6648-01 test method, the test temperature of −12 °C and the stress continuous action of 100 g weight for 240 s were selected to study the low-temperature creep properties of composite inorganic powder-modified asphalt under different types and dosages. At the same time, two test temperatures of −18 and −24 °C and 100 g weight were selected to apply stress for 60 s. To thoroughly assess the low-temperature performance of GTCM-modified asphalt, the bending creep test of several types of GTCM-modified asphalt was conducted. The law of change with temperature. During the test, tourmaline-modified asphalt, mineral powder asphalt mastic and matrix asphalt were used as control tests. The specific test scheme is shown in the dynamic shear rheological performance test method.

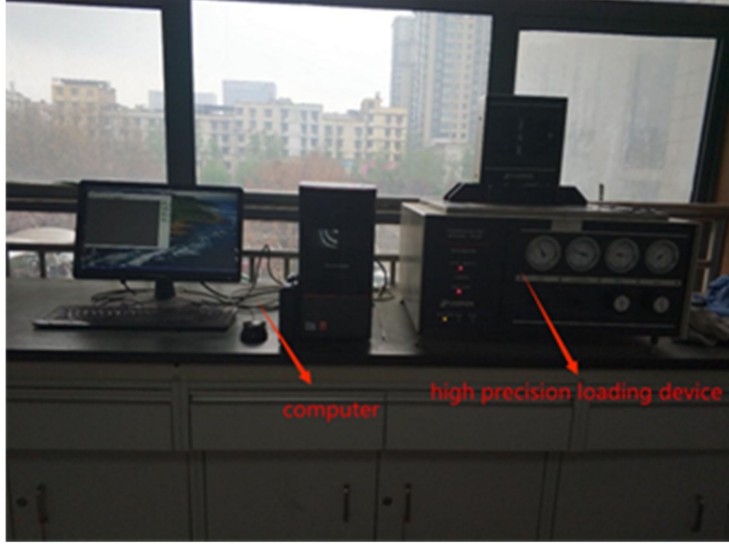

**Figure 5.** Asphalt low temperature bending beam rheometer.

*2.5. FTIR Spectral Analysis Method*

The infrared spectra of samples were measured by TENSOR27 FTIR Fourier transform infrared spectrometer produced by Thermo Nicolet Corporation from Madison, WI, USA, as shown in Figure 6. The instrument can measure the spectral range of 8300~30 cm$^{-1}$. SNR better than 40,000: 1 (peak–peak); resolution less than or equal to 4 cm$^{-1}$; the test site is the Key Laboratory of Applied Surface and Colloidal Chemistry, College of Chemical Engineering, Shaanxi Normal University, Xi'an, China.

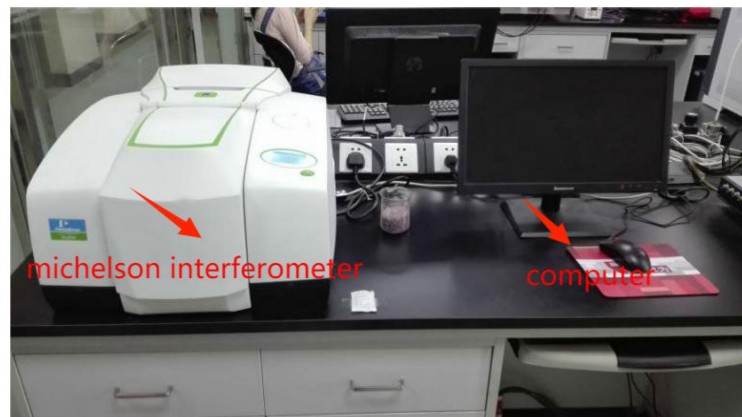

**Figure 6.** Fourier transform infrared spectrometer.

## 3. Result and Discussion

*3.1. Basic Performance*

To clarify the content of different types of GTCM in asphalt and its influence on asphalt performance, inorganic micropowder modified asphalt with different contents has been prepared, and a thorough investigation of the temperature sensitivity, high temperature, and anti-aging characteristics of modified asphalt. Table 4 presents the outcomes.

**Table 4.** Main technical indexes of matrix asphalt.

| Test Project | | Unit | Test Results |
|---|---|---|---|
| Penetration (25 °C) | | 0.1 mm | 64 |
| Penetration index PI | | - | −1.37 |
| Ductility (10 °C) | | cm | 38.3 |
| Softening point (R&B) | | °C | 47.9 |
| 60 °C Kinetic viscosity | | Pa·s | 226 |
| After TFOT | Quality change | % | 0.12 |
| | Penetration ratio | % | 74.1 |
| | Ductility (10 °C) | cm | 12.4 |

According to Table 5.

(1) The temperature sensitivity of GTCM-modified asphalt is superior to tourmaline micropowder-modified asphalt under the same circumstances, and the improvement impact is more pronounced the more graphene micropowder is present. Compared to tourmaline micropowder modified asphalt, the temperature sensitivity of composite micropowder modified asphalt is increased by up to 87.2%.

(2) The high-temperature stability of asphalt changed with GTCM is superior to asphalt modified with tourmaline micropowder under the same circumstances. The improvement effect is more significant with higher graphene micropowder content. Compared to tourmaline micropowder modified asphalt, the GTCM-modified asphalt showed a 6.35% improvement in 25 °C penetration, 7.62% improvement in softening point, and 7.17% improvement in equivalent softening point.

(3) Under the same circumstances, the anti-aging performance of asphalt modified with composite micropowder is superior to that of asphalt modified with tourmaline micropowder. Additionally, the higher the content of graphene micropowder, the more significant the improvement effect. In comparison to tourmaline micropowder modified asphalt, the GTCM-modified asphalt showed an improvement of 8.60% in 25 °C penetration residual ratio and 57.58% in softening point increment.

**Table 5.** Basic performance test results of inorganic powder modified asphalt.

| Asphalt Type | Dosage (%) | Needle Penetration (100 g, 5 s, 0.1 mm) | | | | | Residual Needle Penetration Ratio (%) | Softening Point (°C) | | | |
|---|---|---|---|---|---|---|---|---|---|---|---|
| | | Before Aging | | | | After Aging | | Before Aging | Equivalent Softening Point $T_{800}$ | After Aging | Increment |
| | | 15 °C | 25 °C | 30 °C | PI | 25 °C | | | | | |
| Matrix asphalt | - | 25 | 75 | 138 | −1.317 | 43 | 57.2 | 41.5 | 45.5 | 49.5 | 8.0 |
| T asphalt | 10 | 23 | 65 | 116 | −1.031 | 40 | 61.8 | 49.0 | 48.0 | 52.0 | 3.0 |
| | 20 | 22 | 61 | 107 | −0.909 | 39 | 63.7 | 51.0 | 49. | 52.5 | 1.5 |
| | 30 | 20 | 58 | 99 | −0.895 | 38 | 64.3 | 52.5 | 50.0 | 53.5 | 1.0 |
| GTCM-0.5 | 10 | 24 | 64 | 115 | −0.826 | 41 | 63.7 | 49.5 | 49.0 | 52.0 | 2.5 |
| | 20 | 22 | 61 | 103 | −0.669 | 40 | 65.3 | 52.0 | 50.0 | 53.0 | 1.0 |
| | 30 | 21 | 57 | 95 | −0.536 | 38 | 67.6 | 53.0 | 51.5 | 54.5 | 1.5 |
| GTCM-1.0 | 10 | 24 | 63 | 109 | −0.581 | 42 | 66.2 | 50.5 | 50.0 | 52.5 | 2.0 |
| | 20 | 23 | 60 | 98 | −0.369 | 40 | 66.9 | 52.5 | 51.5 | 53.5 | 1.0 |
| | 30 | 22 | 55 | 91 | −0.244 | 40 | 71.3 | 55.5 | 53.0 | 56.5 | 1.0 |
| GTCM-1.5 | 10 | 24 | 62 | 107 | −0.446 | 42 | 67.6 | 51.5 | 50.5 | 53.0 | 1.5 |
| | 20 | 24 | 59 | 100 | −0.228 | 41 | 69.0 | 53.0 | 52.0 | 54.0 | 1.0 |
| | 30 | 23 | 55 | 93 | −0.115 | 40 | 72.9 | 56.5 | 53.5 | 57.5 | 1.0 |

The high-temperature stability of asphalt changed with GTCM is superior to asphalt modified with tourmaline micropowder under the same circumstances, that is, GTCM-modified asphalt had much better temperature sensitivity, high-temperature resistance, and anti-aging capabilities than tourmaline-modified asphalt, and as the amount of graphene micropowder rose, the improving impact steadily grew, which has been proved in the existing research [21,30,31].

*3.2. Dynamic Shear Rheological Property Analysis*

3.2.1. Temperature Scanning

The temperature scanning test of GTCM-modified asphalt was carried out. The outcomes are displayed in Figures 7–10, and the rheological properties of GTCM-modified asphalt in the continuous temperature range are analyzed.

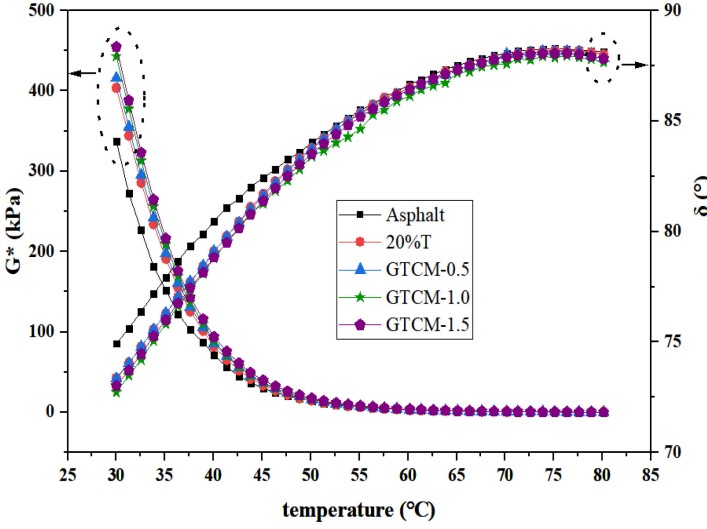

**Figure 7.** Curves of complex shear modulus and phase angle of different types of GTCM-modified asphalt with temperature.

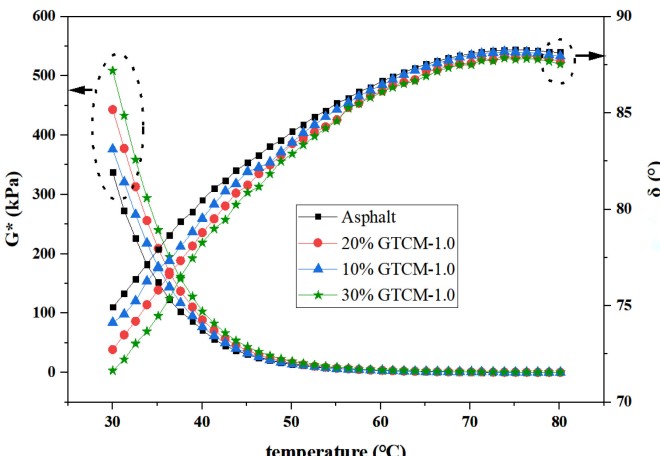

**Figure 8.** Curves of complex shear modulus and phase angle of GTCM with different contents changing with temperature.

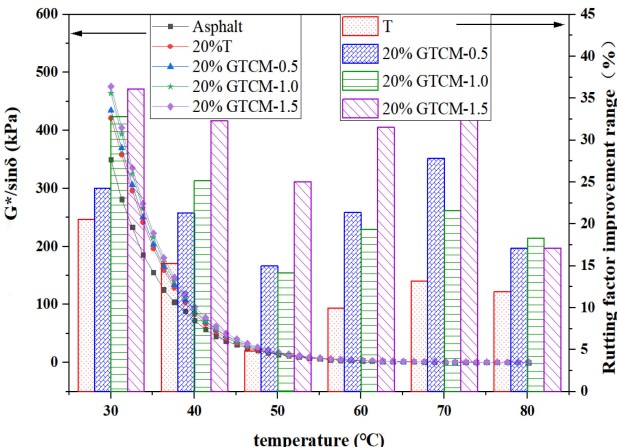

**Figure 9.** Rutting factor of different types of GTCM-modified asphalt and its improvement trend with temperature.

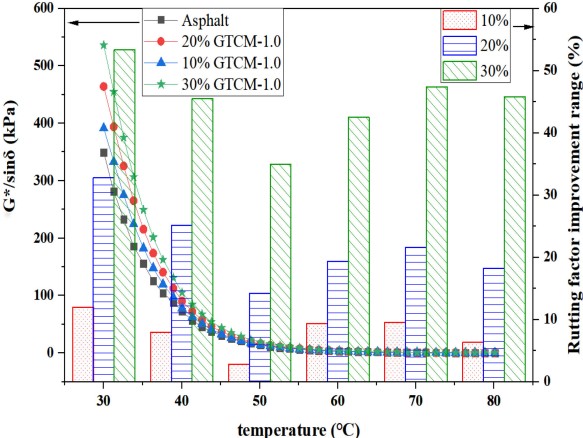

**Figure 10.** Rutting factor of different types of GTCM-modified asphalt and its improvement trend with temperature.

It is shown in Figure 7.

(1) Similar to matrix asphalt and tourmaline micropowder modified asphalt, the complicated shear modulus change trend of various types of GTCM-modified asphalt is the temperature under the same dose. Temperature increases cause an exponential

decline in the complex modulus, indicating that the GTCM is used for modified asphalt. Asphalt's viscoelastic characteristics remained the same, but the complex modulus of different types of GTCM-modified asphalt at the same temperature was higher than that of tourmaline micropowder modified asphalt and matrix asphalt.

(2)　Similar to matrix asphalt and tourmaline-modified asphalt, the phase angle of GTCM-modified asphalt has the same content. Additionally, the phase angle increases with temperature. During dynamic mechanical testing, the phase angle δ of stress and strain hysteresis indicates the ratio of viscosity and elasticity of viscoelastic materials. The phase angle of completely elastic material is 0°, and the phase angle of completely viscous material is 90°, while the phase angle of viscoelastic material is in the range of 0–90°. The phase angle of various asphalt kinds is within 88° as the temperature rises from 30 °C to 80 °C. It also shows that the addition of GTCM to asphalt does not change the viscoelastic properties of asphalt.

It is shown in Figure 8.

(1)　The complex shear modulus and phase angle fluctuation trends of modified asphalt varying GTCM concentrations and temperature are essentially identical to those of matrix asphalt. The temperature rises cause an exponential decline in the complex modulus and an increase in the phase angle. It shows that the viscoelastic properties of asphalt are not changed after the GTCM material is used for modified asphalt in the range of 30%.

(2)　The complex modulus between matrix asphalt and GTCM-modified asphalt is noticeably different in the region of 30 to 50 °C, and the higher the content, the higher the complex shear modulus. When the temperature exceeds 50 °C, the content has less of an impact on the complex modulus of GTCM-modified asphalt, and the complex modulus between different contents is very small, but the overall trend is still 30%. The complex modulus of GTCM-modified asphalt is the largest.

(3)　In the range of 30~55 °C, the phase angle between different content of GTCM-modified asphalt and matrix asphalt is obvious, and the larger the content, the smaller the phase angle. When the temperature exceeds 55 °C, the content has less of an impact on the phase angle of GTCM-modified asphalt. The phase angle between different content composite modified asphalt is small, but the overall trend is still 30%. The phase angle of GTCM-modified asphalt is the smallest.

It is shown in Figure 9.

(1)　Under the condition of the same content, the rutting factor of different kinds of GTCM is similar to that of tourmaline-modified asphalt and matrix asphalt, which falls down dramatically as the temperature rises, but the rutting coefficient of GTCM to asphalt is larger than that of tourmaline and matrix asphalt, indicating that the rutting factor of GTCM to asphalt is better than that of tourmaline composite powder.

(2)　According to the results, between 30 and 50 °C, the rutting factors of matrix asphalt and modified asphalt are very different, while the rutting factors of different types of GTCM-modified asphalt are small. In the temperature range of more than 50 °C, the rut factor of various types of asphalt has little difference and is at a low level.

(3)　When the temperature rises, the improvement of the rutting factor decreases first, then increases, and finally increases again. At different temperatures, the improvement of the rutting coefficient of different types of GTCM-modified asphalt is different. Below 50 °C, with the addition of GTCM, the improvement of the rutting factor is gradually increasing. In the temperature range of 50~70 °C, with the addition of graphene powder, the improvement of the rutting factor shows a trend of decreasing first and then increasing. At the temperature of 50~70 °C, with the addition of graphene powder, the improvement degree of the rutting factor shows a trend of increasing first and then decreasing, indicating that GTCM-1.5-modified asphalt has the best anti-rutting ability when the temperature does not exceed 70 °C, while GTCM-1.0 has the best anti-rutting ability at the temperature of 70~80 °C.

It is shown in Figure 10.

(1) The change trend of the rut factor of GTCM-modified asphalt with temperature is basically the same as that of matrix asphalt, and the rut factor decreases exponentially with the increase in temperature. The rutting factor of modified asphalt increases with increasing GTCM content at the same temperature, suggesting that adding more GTCM can help asphalt become more resistant to rutting.

(2) The improvement range of the rutting factor of modified asphalt with different content of GTCM is basically the same as that of temperature, which decreases first, then increases and then increases with the increase in temperature. At the same temperature, the improvement range of the rutting factor of modified asphalt with GTCM increases with the increase in content.

(3) At 60 °C, the rut factor of matrix asphalt is 2.97 kPa, and the rut factor of GTCM-modified asphalt is 3.25 kPa at 10% dosage, which is 9.43% higher than that of matrix asphalt. In comparison to matrix asphalt, the rutting factor of GTCM-modified asphalt with a 20% component is 19.53% greater at 3.55 kPa. The 30% GTCM-modified asphalt has a rutting factor of 4.23 kPa, which is 42.42% more than matrix asphalt.

### 3.2.2. Frequency Scanning Results and Analysis

To clarify the viscoelastic properties of GTCM-modified asphalt under different load frequencies, it is necessary to study the rheological properties of GTCM-modified asphalt under different load frequencies. In Figures 11–13, the precise test results are displayed.

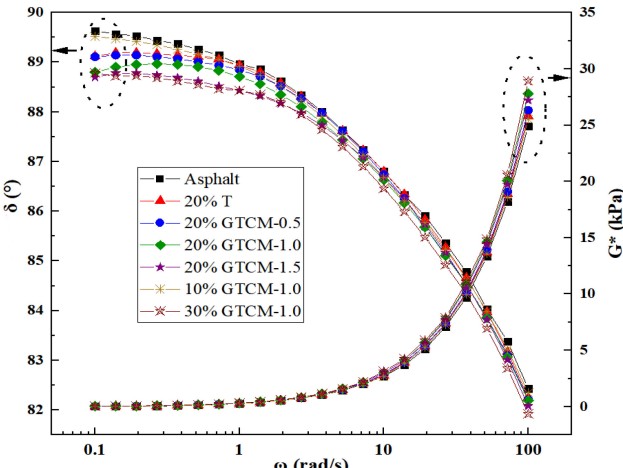

**Figure 11.** Curves of complex modulus and phase angle of GTCM-modified asphalt with frequency under different types and contents.

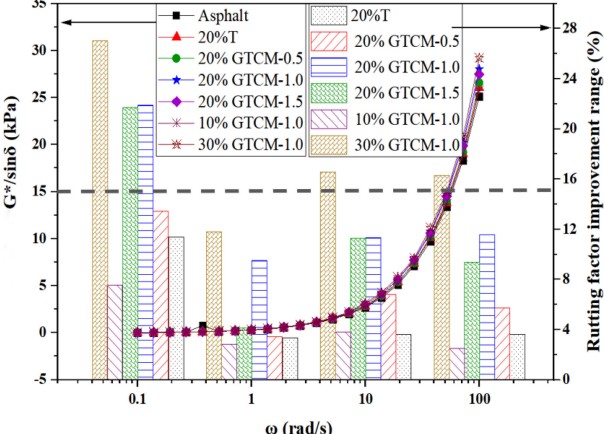

**Figure 12.** Rutting factor of GTCM-modified asphalt and its improvement trend with frequency under different types and contents.

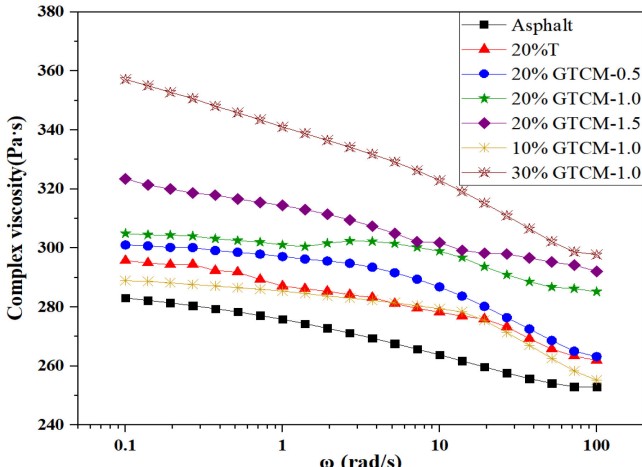

**Figure 13.** The curve of complex viscosity of GTCM-modified asphalt with frequency under different types and contents.

It is shown in Figure 11.

(1) The complex shear modulus and phase angle fluctuation trends of various types of GTCM-modified asphalt with frequency are essentially identical to those of matrix asphalt and tourmaline micropowder-modified asphalt. The complex modulus increases exponentially with the increase of frequency, and the phase angle decreases gradually with the increase of frequency, indicating that the viscoelastic properties of asphalt are not changed after the GTCM is used for modified asphalt. However, the phase angle of various types of GTCM-modified asphalt is smaller than that of tourmaline micropowder-modified asphalt and matrix asphalt, and the complex modulus of these modified asphalt types is larger at the same frequency.

(2) The complex shear modulus and phase angle of GTCM-modified asphalt has the same trend with frequency under the same type and different content. At the same frequency, the complex shear modulus increases with increasing GTCM amount while the phase angle decreases.

It is shown in Figure 12.

(1) Under the same dosage, the rutting factor of different types of GTCM-modified asphalt increases exponentially with the increase of frequency, which is basically consistent with the trend of the rutting factor of tourmaline micropowder-modified asphalt and matrix asphalt with frequency. However, under the same conditions, the finding that matrix asphalt and tourmaline micropowder-modified asphalt have lower rutting factors than GTCM-modified asphalt suggests that GTCM has the best impact on enhancing the rutting resistance of asphalt.

(2) The rut factor of GTCM-modified asphalt with the same type and different content is basically the same as the frequency change trend, it gets bigger when the frequency gets bigger. At the same frequency, the larger the content of GTCM, the greater the rut factor, indicating that increasing the content helps to improve the rut resistance of GTCM-modified asphalt.

(3) When the frequency is 10 rad/s, the rut factor of tourmaline micropowder modified asphalt is 2.78 kPa. The rut factors of GTCM-0.5, GTCM-1.0 and GTCM-1.5-modified asphalt are 2.87, 3.00 and 2.99 kPa, respectively, which are 3.24%, 7.91% and 7.55% higher than that of tourmaline micropowder modified asphalt.

(4) When the frequency is 10 rad/s, the rut factor of the matrix asphalt is 2.69 kPa, and the rut factors of the graphene tourmaline composite powder-modified asphalt at 10%, 20%, and 30% are 2.79, 3.00, and 3.14 kPa, respectively. Compared with the matrix asphalt, it increased by 3.72%, 14.20%, and 16.73%, respectively.

It is shown in Figure 13.

(1)  Under various kinds and contents, the complex viscosity of GTCM-modified asphalt falls as frequency increases. When compared to matrix asphalt, GTCM-modified asphalt has a greater complex viscosity, and is higher than that of tourmaline micropowder-modified asphalt at the same content, indicating that the shear performance of GTCM-modified asphalt is better than that of tourmaline micropowder-modified asphalt matrix asphalt.

(2)  At the same frequency, by comparing the complex viscosity of different types of GTCM-modified asphalt, It can be shown that the complex viscosity of composite-modified asphalt increases with increasing graphene micropowder amount, that is, the shear resistance of GTCM-1.5-modified asphalt is the best.

(3)  When the frequency is 10 rad/s, the complex viscosity of tourmaline micropowder modified asphalt is 278.28 Pa·s. The complex viscosity of GTCM-0.5, GTCM-1.0 and GTCM-1.5-modified asphalt is 286.81, 298.92 and 301.85 Pa·s, respectively, which is 3.07%, 7.42% and 8.46% higher than that of tourmaline micropowder modified asphalt.

(4)  When the frequency is 10 rad/s, the complex viscosity of the matrix asphalt is 263.81 Pa·s. The complex viscosity of the GTCM-modified asphalt at 10%, 20% and 30% is 279.42, 298.92 and 322.9 Pa·s, respectively, which is 5.92%, 13.31% and 22.40% higher than that of the matrix asphalt.

Existing studies have shown that GO can improve the rutting resistance of 90 A and SBS MA in the temperature range of 30–80 °C [16]. The existing research shows that tourmaline can significantly improve the high-temperature rheological properties of asphalt. The dynamic shear rheological properties of tourmaline-modified asphalt steadily improve with increasing tourmaline amount. When the tourmaline content exceeds 14%, The effect of tourmaline content on the enhancement of asphalt's high-temperature performance is insignificant [32]. Through comprehensive comparative analysis of the rheological properties of GTCM-modified asphalt at different temperatures and frequencies, it can be concluded that the anti-rutting performance of GTCM-modified asphalt is better than that of tourmaline micropowder-modified asphalt. Compared with the same content of tourmaline micropowder-modified asphalt, The anti-rutting performance of GTCM-0.5, GTCM-1.0 and GTCM-1.5-modified asphalt can be increased by 12.95%, 10.12% and 24.25%, respectively.

### 3.3. Analysis of Rheological Properties at Low Temperature

The low-temperature creep tests of GTCM-modified asphalt with different types and contents were carried out at −12, −18 and −24 °C by low-temperature bending beam rheometer. The test outcomes are displayed in Figures 14–16.

(1) The stiffness modulus (−12 °C) of GTCM-modified asphalt under different types and contents.

It is shown in Figure 14.

Under various kinds and doses, the creep stiffness modulus of GTCM-modified asphalt falls as loading time increases, as with matrix asphalt, tourmaline micropowder-modified asphalt, and mineral powder asphalt mastic, the change pattern is essentially the same. However, during the 240 s loading period, the stiffness modulus of graphene micropowder reinforced tourmaline-modified asphalt is smaller than that of tourmaline micropowder-modified asphalt and mineral powder asphalt mastic, indicating that the low-temperature performance of GTCM-modified asphalt is better than that of tourmaline micropowder modified asphalt.

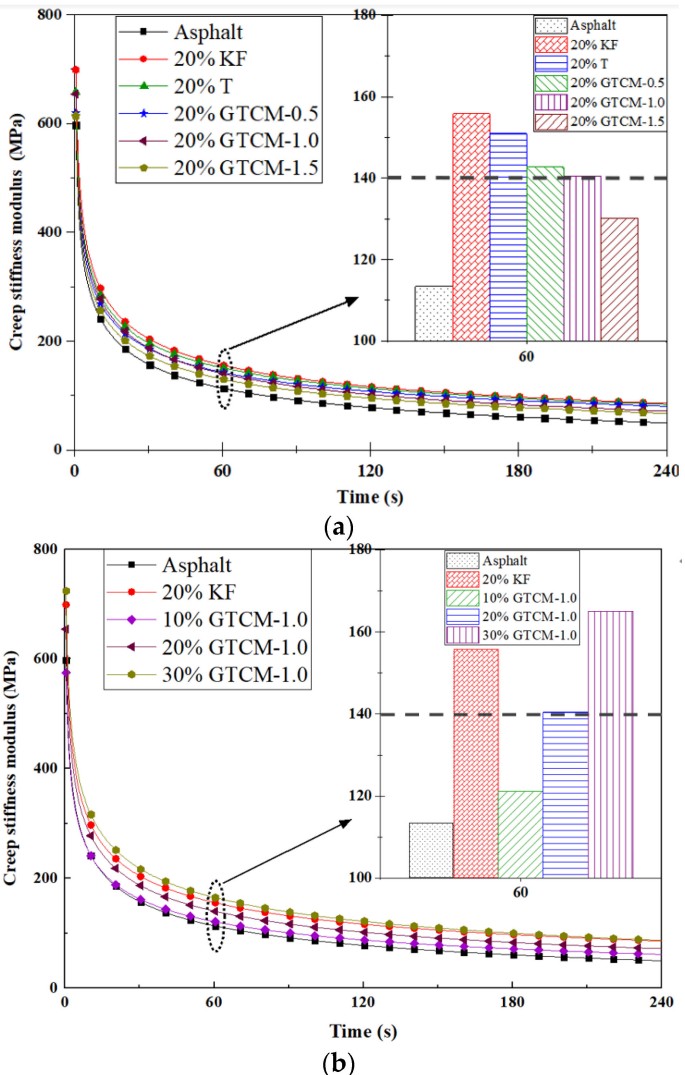

**Figure 14.** Stiffness modulus of GTCM-modified asphalt with different types and contents. (**a**) Different kinds. (**b**) Different dosage.

The creep stiffness modulus of GTCM-modified asphalt with the same type and different content also decreases with the extension of load action time. The different content only changes the creep stiffness modulus of asphalt and does not affect the rheological properties of asphalt under low-temperature load. The larger the content of GTCM, the larger the creep stiffness modulus of modified asphalt, indicating that too much content will adversely affect the low-temperature performance of asphalt.

When the loading time is 60 s, the creep stiffness modulus of tourmaline powder modified asphalt is 150.95 MPa, and the creep stiffness modulus of GTCM-0.5-modified asphalt is 142.74 MPa, which is 5.75% higher than that of tourmaline powder modified asphalt. The creep stiffness modulus of GTCM-1.0-modified asphalt is 140.43 MPa, which is 6.97% higher than that of tourmaline powder-modified asphalt. The creep stiffness modulus of GTCM-1.5-modified asphalt is 130.22 MPa, which is 13.73% higher than that of tourmaline powder modified asphalt, that is when the loading time is 60 s, the low-temperature performance of GTCM-1.5-modified asphalt is the best.

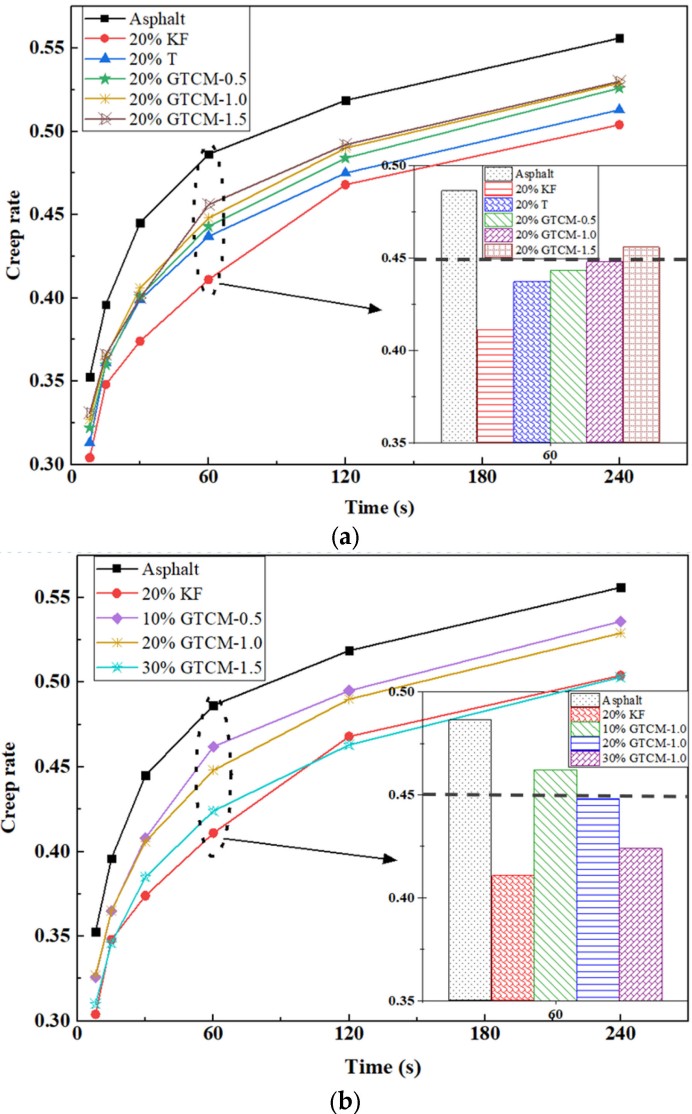

**Figure 15.** Creep rate of GTCM-modified asphalt under different kinds and contents. (**a**) Different kinds. (**b**) Different dosage.

(2) Creep rate ($-12$ °C) of GTCM-modified asphalt under different types and contents
It is shown in Figure 15.

Under different types and contents, the creep rate of GTCM-modified asphalt gradually increases with the extension of load time, and its change trend is basically the same as that of matrix asphalt, tourmaline micropowder-modified asphalt and mineral powder asphalt mastic. However, in the 240 s load period, the creep rate of GTCM-modified asphalt is greater than that of tourmaline micropowder-modified asphalt and mineral powder asphalt mastic, indicating that the low-temperature performance of GTCM-modified asphalt is better than that of tourmaline micropowder modified asphalt.

The creep rate of GTCM-modified asphalt with the same type and different content also increases with the extension of load action time. The different content only changes the creep rate of asphalt and does not affect the rheological properties of asphalt under low temperature load. The larger the content of GTCM, the smaller the creep rate of modified asphalt, indicating that too much content will have an adverse effect on the low-temperature performance of modified asphalt.

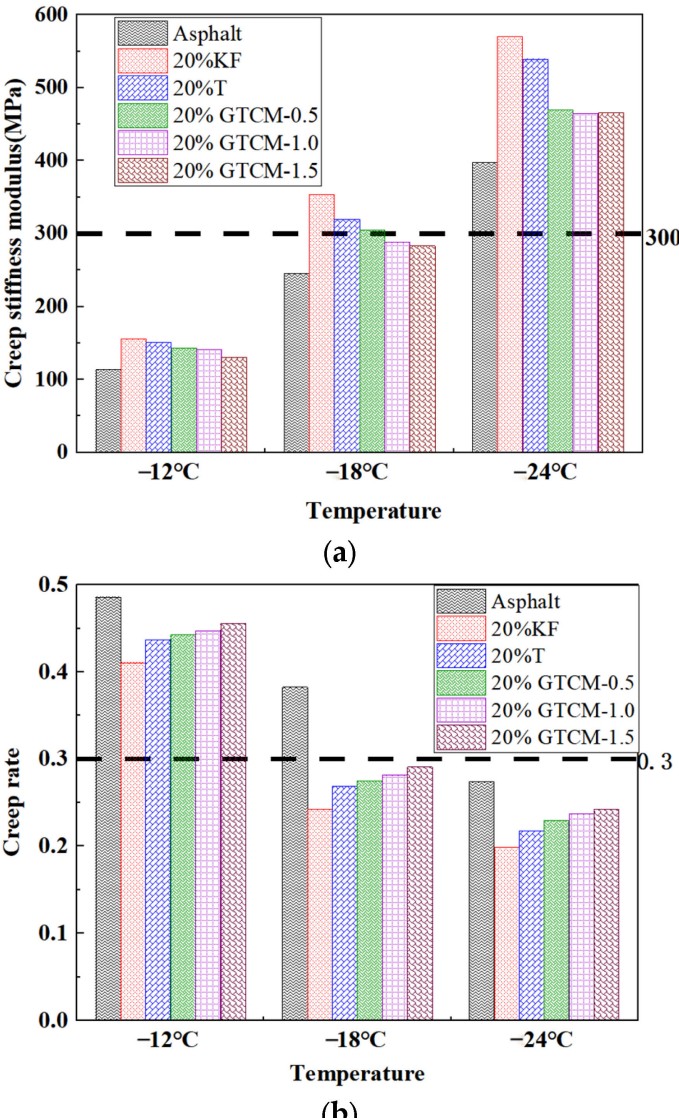

**Figure 16.** Creep stiffness modulus and creep rate of GTCM-modified asphalt at different temperatures, (**a**) Creep stiffness modulus and (**b**) Creep rate.

When the loading time is 60 s, the creep rate of tourmaline powder-modified asphalt is 0.437, and the creep rate of GTCM-0.5-modified asphalt is 0.443, which is 1.37% higher than that of tourmaline powder-modified asphalt. The creep rate of GTCM-1.0-modified asphalt is 0.448 MPa, which is 2.52% higher than that of tourmaline powder-modified asphalt. The creep rate of GTCM-1.5-modified asphalt is 0.456, which is 4.35% higher than that of tourmaline powder-modified asphalt. When the loading time is 60 s, the low-temperature performance of GTCM-1.5-modified asphalt is the best.

(3) The stiffness modulus and creep rate of GTCM-modified asphalt at different temperatures (60 s)

It can be clearly seen from Figure 16.

With a drop in temperature, the creep stiffness modulus of GTCM-modified asphalt steadily rises. At the same temperature, although the creep stiffness modulus of GTCM-modified asphalt is higher than that of matrix asphalt, it is lower than the creep stiffness modulus of tourmaline micropowder-modified asphalt and mineral powder asphalt mastic under the same dosage.

With a drop in temperature, the creep rate of GTCM-modified asphalt steadily reduces. The creep rate of GTCM-modified asphalt is higher than that of tourmaline micropowder

modified asphalt and mineral powder asphalt mastic at the same dose, yet it is less than that of matrix asphalt at the same temperature.

When the ambient temperature is $-12$ and $-18\ ^\circ$C, the creep stiffness modulus of the GTCM-modified asphalt is lower than 300 MPa, and the creep rate is only greater than 0.3 at $-12\ ^\circ$C, indicating that the GTCM-modified asphalt has good low-temperature crack resistance at $-12\ ^\circ$C.

In summary, the low-temperature crack resistance of GTCM-modified asphalt is better than that of tourmaline micropowder-modified asphalt, and the greater the content of graphene micropowder, the more obvious the improvement effect on low-temperature performance. Relevant research shows that tourmaline powder-modified asphalt has lower flexibility and weaker stress relaxation ability than matrix asphalt. Tourmaline reduces the low-temperature crack resistance of asphalt to a certain extent [33]. Adding graphene oxide can effectively absorb the load. Thus, the flexibility and deformation ability of asphalt are enhanced, and the low-temperature cracking resistance of asphalt binder is improved [21]. The reason may be that the flexibility of graphene is better than that of tourmaline. Because graphene has a far bigger specific surface area than tourmaline, it is distributed throughout asphalt at a considerably higher density than tourmaline, which improves the flexibility of asphalt in low-temperature environments. At the same time, asphalt molecules may partially permeate the graphene lattice because of its stable two-dimensional lamellar structure, and some stress is absorbed by graphene during loading, which weakens the direct effect of stress on asphalt, thereby improving the low-temperature crack resistance of asphalt.

*3.4. Analysis of Fourier Infrared Spectrum*

The infrared spectra of matrix asphalt, tourmaline-modified asphalt, GTCM-0.5-modified asphalt, GTCM-1.0-modified asphalt and GTCM-1.5-modified asphalt were tested, respectively. The test results are shown in Figure 17.

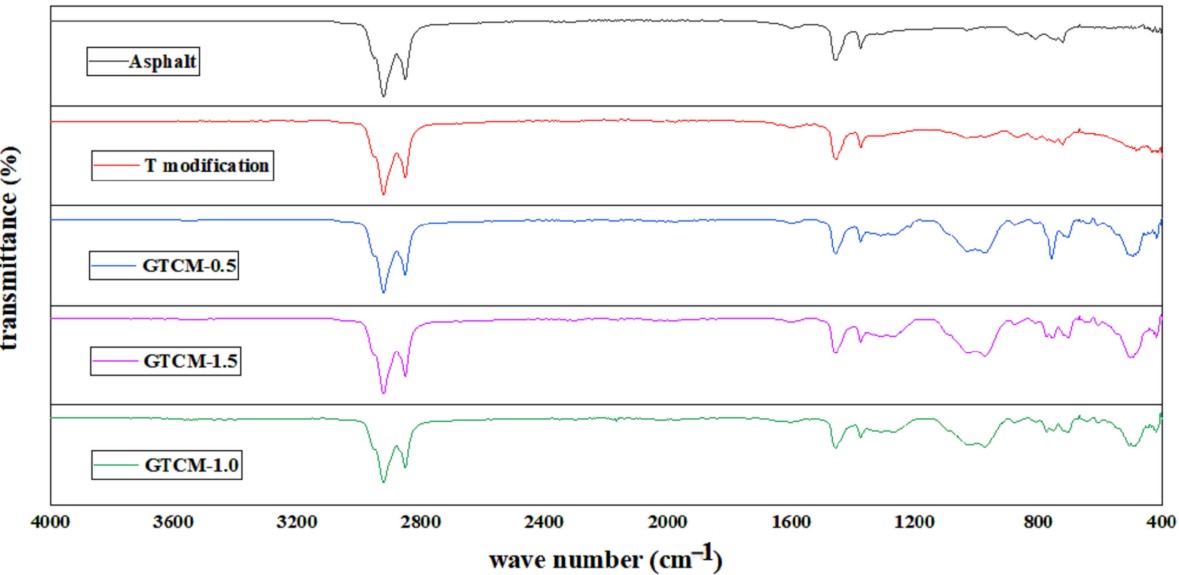

**Figure 17.** FTIR spectrum of GTCM-modified asphalt.

From the FTIR spectra of different types of asphalt in Figure 17, it can be seen that there are some differences in the absorption peaks of different types of asphalt in the infrared spectrum. In the FTIR spectrum of matrix asphalt, there are strong absorption peaks at 2920.05 and 2850.96 cm$^{-1}$, which are antisymmetric and symmetric stretching vibrations of CH$_2$ in long-chain alkanes and cycloalkanes. The absorption peaks of C = C and C–C skeleton vibration in the benzene ring of aromatic compounds were 1601.98, 1456.26 and 1376.34 cm$^{-1}$. The absorption peak of S = O stretching vibration of dibenzyl

sulfoxide at 1031.10 cm$^{-1}$. The four smaller characteristic peaks at 864~721 cm$^{-1}$ are the C–H out-of-plane bending vibration on the benzene ring of aromatic compounds.

The difference between the FTIR spectra of the tourmaline-modified asphalt and the GTCM-modified asphalt and the matrix asphalt mainly appears in the low-frequency region, which is reflected in the absorption peaks with low sharp strength in the range of 720~400 cm$^{-1}$. The tourmaline-modified asphalt has obvious absorption peaks at 483.64, and 433.04 cm$^{-1}$. GTCM-0.5-modified asphalt showed obvious absorption peaks at 704.48, 634.90, 493.69 and 417.97 cm$^{-1}$. The absorption peaks of the low-frequency region in the FTIR spectrum of GTCM-1.0-modified asphalt mainly appeared at 641.50, 504.64, 491.36 and 419.06 cm$^{-1}$. The different absorption peaks of GTCM-1.5-modified asphalt FTIR spectrum are mainly distributed at 704.50, 634.98, 606.84, 505.78 and 418.85 cm$^{-1}$. Compared with the tourmaline FTIR spectrum, the primary source of the low-frequency region's small and sharp absorption peaks is the lattice vibration of some oxides such as $ZnO$, $Fe_2O_3$, $Fe_3O_4$, $Al_2O_3$, $MgO$, $TiO_2$, and $SiO_2$. This part of the absorption peak in the modified asphalt FTIR spectrum mainly comes from tourmaline. After adding tourmaline or GTCM material to asphalt, asphalt does not undergo a chemical reaction that would create new functional groups, but only increases the infrared activity of asphalt material in a lower frequency range.

## 4. Conclusions

In this paper, GTCM-modified asphalt is taken as the object. Firstly, the influence of different temperatures and different frequencies on the high-temperature performance of GTCM-modified asphalt is evaluated by dynamic shear rheological test, and the viscoelastic properties of GTCM-modified asphalt under different stress and different temperatures are analyzed. Secondly, the low-temperature rheological properties of GTCM-modified asphalt were analyzed by bending beam rheological test. Finally, the mechanism was analyzed by FTIR. The main conclusions are as follows:

(1) Compared to tourmaline-modified asphalt, GTCM-modified asphalt has much better temperature sensitivity, high temperature resistance, and anti-aging capabilities. The improved impact will also rise when the graphene powder content is increased, and the temperature sensitivity of GTCM-modified asphalt can be increased by up to 87.2%. The improvement of penetration, softening point, and equivalent softening point at 25 °C can be up to 6.35%, 7.62% and 7.17%, respectively. The improvement of penetration residual ratio and softening point increment at 25 °C can be up to 8.60% and 57.58%, respectively.

(2) The viscoelasticity of asphalt was unaffected by the addition of GTCM. The anti-rutting performance of GTCM-modified asphalt is better than that of tourmaline-modified asphalt. At the same temperature, it increases with the increase in composite powder content. At different temperatures, the modification effect of the composite powder is still optimal. Compared with the same amount of tourmaline-modified asphalt, the anti-rutting performance of GTCM-0.5, GTCM-1.0 and GTCM-1.5-modified asphalt can be improved by 12.95%, 10.12% and 24.25%, respectively.

(3) Under various load frequencies, GTCM-modified asphalt with various contents performs better than tourmaline-modified asphalt in terms of anti-rutting and shear resistance, and its performance improves as the amount of graphene powder and GTCM rises. When the load was 60 s, the creep stiffness modulus of GTCM-0.5, GTCM-1.0 and GTCM-1.5-modified asphalt decreased by 5.75%, 6.97% and 13.73%, respectively, and the creep rate increased by 1.37%, 2.52% and 4.35%, respectively.

(4) GTCM has stronger low-temperature crack resistance than tourmaline, and the improvement in low-temperature performance is increasingly noticeable as graphene powder gets larger. However, the high content of GTCM will have a negative impact on its low-temperature performance. When the ambient temperature is 12 °C, GTCM-modified asphalt has good low-temperature crack resistance.

(5) FTIR analysis showed that after adding GTCM material or tourmaline to the matrix asphalt, no new functional groups were produced due to the chemical reaction with the asphalt.

**Author Contributions:** T.G.: Conceptualization, Project administration, Supervision, Writing-Review and Editing. H.C.: Conceptualization, Formal analysis, Methodology, Visualization. D.T.: Supervision, Project administration, Data curation, Formal analysis. S.D.: Conceptualization, Writing-Original Draft, Supervision, Investigation. C.W.: Conceptualization, Supervision, Writing-Original Draft. D.W.: Conceptualization, Project administration, Supervision, Investigation. Y.C.: Funding acquisition, Investigation. Z.L.: Project administration, Resources. All authors have read and agreed to the published version of the manuscript.

**Funding:** This research was sponsored by the Henan Province 2021 Science and Technology Key Project 'Performance and Synergistic Mechanism of Graphene/Tourmaline Composite Modified Asphalt' (Grant No. 212102310089), North China University of Water Resources and Hydropower Research Project (Grant No. 201811035) and Fundamental Research Funds for the Central Universities (Grant No. 300102219314). That sponsorship and interest are gratefully acknowledged. Open Fund Project of Henan Province Engineering Technology Research Center for Environment-friendly and High-performance Pavement Materials (Henan Science foundation (2020) No.18).

**Institutional Review Board Statement:** Not applicable.

**Informed Consent Statement:** Not applicable.

**Data Availability Statement:** Not applicable.

**Conflicts of Interest:** The authors declare no conflict of interest.

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
