# Peer review of "Rheological Properties of Composite Inorganic Micropowder Asphalt Mastic"

_coatings, doi:10.3390/coatings13061068_

Round 1
Reviewer 1 Report
Dear authors
Appreciating your good writing, but there are some points and flaws that are mentioned.
-The use of the word [mortar] is less common in the technical terminology of paving, the word [mastic] is more often used, and it is suggested to replace it.
-How is the last column[increment] calculated in Table 3? Are they accurate? What is the unit? Please check.
- In Figure 1, write the letters A, B, and C in the pictures.
-The linear flowchart in Figure 1 has nothing new to say, and the same content is mentioned in the text, so it should be removed.
- Put the physical properties of all three substances in a table.
At the beginning, many paragraphs are numbered, when it is not necessary and each paragraph covers a topic by itself.
- In Figure 8, why does it first [G/sind ] decrease by 20 GTCM-0.5 and then increase? Please add more details.
- In Figure 14, [different kinds] and in the text image [different types], these two words should be the same.
-The address of some of the references at the end of the article is not complete, as an example, see references number 2 and 3.
With respect.
The writing of the English language of the article is appropriate and understandable for the reader.
Reviewer 2 Report
1. The authors have only cited certain people from certain ethnicity in the paper. This is not ethical when you are trying to publish in an international journal. Please revise ALL the citations and cite researchers from different countries.
2. Please compare the results with other studies. The manuscript only presents some data and is written point by point which is weird and is not compared to other studies.
3. Please enhance the quality of the images some of the have different aspect ratios.
Enhance the English language
Reviewer 3 Report
coatings
Manuscript ID: coatings-2419299
Rheological properties of composite inorganic micropowder asphalt mortar
REFEREE’S COMMENTS
This is an interesting paper on a subject that should be of great interest to many readers. For location of comments, see the belows.
- Abstract:
- There is no need to give the introductory level sentences.
- The abstract should be supported quantitatively.
- Introduction:
- The literature review should be strengthened by citing the papers already available in the literature. For example, see the papers; Transportation Geotechnics 23, 100342; Environmental Earth Sciences 75, 1-9; Resources, Conservation and Recycling 155, 104679; Journal of Materials in Civil Engineering 26 (2), 349-357 ; Journal of hazardous materials 321, 547-556; Journal of Materials in Civil Engineering 30 (8), 04018177; Journal of Materials in Civil Engineering 30 (8), 04018193; and many others.
- Last paragraph: The author(s) should clearly indicate the originality/novelty of her/his/their research.
- Materials and tests:
- Give the full name before using its abbreviation.
- Figure 1; An SEM would be very useful for potential readers.
- Table 1 needs to be modified to have a better organization.
- Figure 3; components of the machine d better to indicate on the photo.
- Figure 4; components of the machine d better to indicate on the photo.
- Figure 5; components of the machine d better to indicate on the photo.
- Result and discussion:
- In this section, the authors should discuss their finding in the light of the papers already available in the literature. This is particularly very important for a scientific paper. Otherwise, it would be a kind of technical report rather than a scientific submission.
- Conclusions:
- The authors should strengthen the conclusions by referring the quantitative findings.
- In General:
- Light-gray-dotted-gridlines in the plot areas would be very useful for the potential readers in order to follow the changes clearly.
- Literature review should be extended.
- Check out the details of the references cited.
Best regards,
-
Round 2
Reviewer 2 Report
the manuscript can be accepted.
Author Response
It is a great honor to receive your recognition and thank you again for your valuable comments on the quality improvement of the article.
Reviewer 3 Report
coatings
Manuscript ID: 2419299
Rheological properties of composite inorganic micropowder asphalt mastic
by
Guo, Chen, Tang, Ding, Wang, Wang, Chen, Li
REFEREE’S COMMENTS
For location of comments, see the belows.
- Abstract:
- Introductory level sentences (first two) could be compacted or even removed from the text.
- Introduction:
- The citing format through the text is incorrect. For example "Amirkhanian S et al. studied ...", "Haibo Ding et al. prepared ...", and many others in the text.
- Give the full name before using its abbreviation first in the text. For example see the line 112, and line 133.
- Figure 1 and others in the submission: "photo" would be better than "macrostructure", "SEM picture" would be better than "microstructure".
- Figures: it would be better to draw light-gray-dotted-gridlines in the plot areas in orer to see the changes clearly.
- line 272; "80 °C."
- Results and Discussion:
- The authors are recommended to discuss their findings in the light of the papers already available in the literature. Currently, the authors made such discussion using very limited number of papers.
- Conclusion:
- The authors are recommended to start this section with a short introduction.
- In General:
- Literature review could be extended by citing the papers already available in the literature.
- The language was found to be fine.
- Check out the details of the references cited.
Best regards,
-
